# System Potential Estimation with Regard to Digitalization: Main Ideas and Estimation Example

**Alexander Geyda** *,† and **Igor Lysenko** †

St. Petersburg Institute for Informatics and Automation of the Russian Academy of Sciences, St. Petersburg 198188, Russia; ilys@iias.spb.su
* Correspondence: geida@iias.spb.su
† These authors contributed equally to this work.

**Abstract:** The article outlines the main concept and examples of mathematical models needed to estimate system potential and digitalization performance indicators. Such an estimation differs in that it is performed with predictive mathematical models. The purpose of such an estimation is to enable a set of problems of system design and functional design of information technologies to be solved as mathematical problems, predictively and analytically. The hypothesis of the research is that the quality of system functioning in changing conditions can be evaluated analytically, based on predictive mathematical models. We suggested a property of the system potential (or system capability) that describes the effects of the compliance of changing system functioning with changing conditions analytically and predictively. Thus, it describes the performance of the use of information operations to realize functioning in changing conditions. The example includes system's environment graph-theoretic models and system's models regarding IT use to react to changing environments. As a result of the suggested models and methods, the quantitative estimation of system potential regarding information technology use becomes possible depending on the parameters and variables of the problems to be solved. Use cases of problems decision with the use of such indicators include choosing optimal information technology implementation and the synthesis of information operation characteristics.

**Keywords:** system; environments; change; potential; efficiency; dynamic capability; IT performance

## 1. Introduction

The solutions of many modern practical problems that arise for practitioners require the study of systems' reactions to environmental impacts and environmental changes ("changing conditions"). For example, changing conditions may take the form of goal changes, impacts of the environment on system elements, and other changes in the conditions of system functioning. These changes necessarily lead to possible changes in system functioning (to the reactions of the system). Changes in the system and its functioning, activity actualization, and planning are initiated and managed by informational actions. Such information is required for measurements, for state checks, for information transfer, and for prescribing further action instructions, descriptions, prescriptions. Only after such information on activity and its changes is obtained can activity and change be implemented. As a result of the actualization and changes, the functioning shall be better adapted to provide the required effects, different from those that would occur without the use of informational operations and so without reactions to environmental influences as a source of disturbances and the need for changes. The abovementioned practical problems are of different types in different domains. Such problems include the improvement of enterprises and organizations [1], the modernization of critical technologies' provision systems or infrastructures [2,3], the digitalization of enterprises

and economies [4], and society informatization [5,6], as well as other problems that are of crucial importance for the security and socioeconomic development of the country [7,8]. Many such problems can be formalized as problems that involve the improvement of systems [9,10]. Technical systems are considered in this study. They are understood as interconnected complexes of parts that include technical devices. Complex technical systems (CTS) may include parts of other types, in particular groups of people and instructions for performing actions related to various types of relationships among people, in addition to technical devices. Such CTS are classified as technological, organizational, or technical systems, depending on the research objectives [11,12]. The need to improve systems due to changed requirements and influences of the system's environment leads, first, to the need for the CTS staff to perform transitional actions to achieve new, updated goals or to resolve inconsistencies and then to implement these transitional actions. As a result, the CTS and its functioning are improved to achieve, perhaps, updated goals or to improve functioning to achieve the goals in changing conditions. The hypothesis of the research is the following: The quality of CTS functioning in changing conditions (i.e., environmental changes, corresponding information operation initiation and fulfillment, and further changes in CTS and then changed CTS functioning and the final effects of it, obtained as a result of sequences of such changes in the environment and CTS) can be evaluated analytically, based on predictive mathematical models of the suggested property of the system's potential (system's capability) and its indicators. Furthermore, such indicator estimation can be made depending on the variables and parameters in practical problems described above, so such practical problems could be solved as mathematical problems of system potential investigation. The research of CTS functioning in changing conditions and IT use to react to changes are traditionally implemented based on the operational properties of such use [13,14]. The operational properties of the objects under study [15] are an extensive class of properties of various objects, such that these properties characterize the results of the activity with these objects. Information operations are defined as elements of activity whose objectives are to obtain information, not to exchange matter and energy. Information operations are implemented in accordance with certain information technology whose objective is to describe the use of information operations. Unfortunately, the mechanisms of the formation of activity effects and the subsequent formation of operational properties, taking into account the use of information operations, including modern (digital) IT, have not been studied in sufficient detail in order to predict the effects of activity with analytic mathematical models, depending on the variable characteristics of the information and on further noninformation operations used, as mathematical problems of evaluation, analysis, and synthesis. Since environmental changes cause information operations and then lead to changes in noninformation operations but do not directly lead to noninformation effects, it is necessary to develop the concept of information and noninformation action dependencies and the concept of effect manifestation as a result of such dependencies. It is the development of the concept of such dependencies that causes conceptual difficulties on the way to build analytical mathematical models of functioning in changing conditions. The results presented in this study are aimed at bridging the gap between the need to solve abovementioned research problems and other operational properties based on analytical predictive mathematical models and methods [9,10,16,17] and the lack of the necessary concepts and methodology for solving usage problems of information operations in the sense of formalizing them as mathematical problems of estimation, analysis, and planning by indicators of operational properties [18]. The system potential with regard to the use of information operations and thus with regard to digitalization [19,20] is investigated to consider the aforementioned complex of CTS operational properties and to solve the problems specified.

## 2. Concepts and Methods to Estimate System Potential Regarding the Use of Information Operations

Informational operations check states and their compliance with requirements and produce information about actions. Their results are instructions and prescriptions to implement subsequent noninformational operations. Therefore, the CTS study should consider the CTS as it improves because

of systematic changes in the requirements and other environmental influences (i.e., in changing conditions). Such an improvement of CTS functioning in changing conditions requires the use of informational operations, performed in accordance with corresponding IT [21]. The need to perform informational and then transitional operations is caused by changing environmental influences [22] and is typical not only for the complex technical and complex technological systems considered to be an example but also for other systems. This is especially true when studying digitalization in various industries, described by such popular terms as digital production, digital medicine, digital economy, and digital state [4,13,23]. As evidenced by the analysis of digitalization [24–26], the digitalization research is based on the dynamic capabilities, the organizational capabilities of the system use, and the ability of the system and its operating personnel to change functions so that the system better meets the changing conditions, improves, and achieves the changing operation goals [16,27–29] due to regular changes (improvements) provided by information operations. In particular, the results of the relevant research are described in publications in the fields of improvement, strategic planning, and development [30] and the digitalization of the economy [14] as well as industrial [31] and military systems [7]. Such studies are conducted with the use of dynamic, organizational, and strategic capabilities [29]. In the known literature, such capabilities and other operational properties are not evaluated analytically with mathematical models. However, to examine the results of digitalization analytically, these results must be connected with the projected results of the system's functions in which these dynamic capabilities use the necessary conceptual and methodological apparatuses that create analytical models and solve practical problems as mathematical problems. This is primarily due to the novelty of the studied properties of systems that function in the changing conditions and due to the complexities to describe the change in system functioning because of informational and causal dependencies on other operations. The new property of the CTS potential, proposed by us, is an operational property that characterizes the CTS's ability to achieve changing (i.e., actual and possible) goals during operation (in a changing environment). This depends on the characteristics of the "target" and "transition" functions of the CTS, including the informational actions performed to check the state of the CTS and the environment, to develop prescriptions for performing technological operations, to execute instructions, and to use action results. The indicator of system potential is evaluated depending on the composition and characteristics of possible actions of different types that form a set of choices in the problems solved. The CTS's potential property is a complex operational (pragmatic, praxeological) property of the system' — that is, a property that describes the results of the system's functions and the results' compliance with requirements in different changing conditions of the environment [32]. The complexity of the systems' potential is caused by the following: the complexity of the description of actions in achieving the activities with the system, the complexity of dependencies at the boundary of the system and the environment in different conditions, the complexity of the description of the goals of the system, and the complexity of the description of activities involved in improving the system. The results obtained in the study of the system potential [19,22] are used in the study of other operational properties of systems that operate under changing conditions and under changing functioning as a result of informational and consequently other operations. They allow practical problems of changing (improved) functioning to be solved as corresponding mathematical problems. Among such problems are digitalization problems, transformation problems by industries, and strategic planning problems. The system's potential is represented as a multidimensional random variable describing measures of system states' compliance with requirements, caused by environmental states. Such interrelated random states of CTS and the environment are modeled based on graph-theoretic interaction models of different kinds (parametric, functional, computational graphs), which allows the computation of multidimensional arrays of states and their probabilities and associated random values. Interactions are modeled based on the concept of the use of information operations for states' causal relationships and based on the concept that such causal relationships are known and can be modeled as functional (computational) dependencies in multidimensional arrays built with the use of graph-theoretic models of functioning in changing conditions.

## 3. Example of Modeling a System's Change Due to Its Environmental Change

Let us consider a simple example of modeling a system's change due to its environmental change. The example consists of a system environment model, which produces possible vectors of states required by the environment and the probabilities (possibilities) for such vectors to be demanded (required) by the environment. One of these vectors shall be realized as a result of environment functioning. Such vectors and probabilities of their realization can be represented as random vectors. Each possible vector required by the system's environmental states leads to a separate model of system functioning under changing conditions. It describes the functioning and changes of the system, including transition operations. Such functioning and system changes (transitions) correspond to the chosen vector of required states. Transition realization requires information operations. The system model allows the effects of operations to be estimated, including transition operations and information operations and their correspondence to environmental requirements, according to the chosen vector of required states. Such a measure is a probabilistic measure estimated for each required state of the vector and can be represented as a discrete vector of correspondences (each correspondence is an element of a vector of probabilistic measures). All possible measures of correspondence (for all required environmental vectors of states) can be represented as a multidimensional random vector or its characteristics — for example, mean, mode, and median. Such a random vector or its characteristics may serve as a system's potential indicator. The indicator varies depending on the capabilities and technologies used to react to the system's environmental changes. IT is one (and necessary required) among such technologies. A measure of the distance between a system's potential indicator value for nondigital ("old") IT used and digital ("new") IT used as a result of digitalization can serve as a digitalization performance indicator.

### 3.1. Complex Technical System Functioning Model

The possible result of using the CTS model are sets $\mathbf{S}_j^s$ of possible CTS states $\mathbf{S}_j := \{s_{uj}, u = \overline{1, U}\}$ for $T_j$. In addition, the result of model use are measures $w_{uj}$ and possibilities $p_{uj}$ for CTS states $s_{uj}$ to be realized at $T_j$. Measures $w_{uj} := p(s_{uj} \sim S_{ij}^d)$ for CTS environment requirements and CTS states correspond according to $\mathbf{S}_q^r$ at $T_j$, that is - $S_{qij}^r$ at the same moment $T_j$ ($S_{ij}^d$)) when $s_{uj}$ is counted. Here "$\sim$" shall take the form of one of the inequality/equality operations, i.e., ">,=,<" and their combinations. The example model $M_{qa}$ of the CTS is calculated for one vector $\mathbf{S}_q^r$ of required states and one set $I_a$ of possible information operation characteristics from the set of possible characteristics of technological information operations $\mathbf{I} = \{I_a, a = \overline{1, A}\}$, depending on one of $A$ IT used. $\mathbf{S}_q^r = < S_{qij}^r >$ is taken from the set of possible sequences of states $\mathbf{S}_q^r$, required by the environment. This vector is $\mathbf{S}_9^r$ in the example shown: $\mathbf{S}_9^r = \mathbf{S}_{23} := < S_{00}^d, S_{11}^d, S_{22}^d, S_{23}^d >$; It means that during the first information operation at moment $T_1$, system functioning must be adapted (conversion of the system shall be performed) to achieve goal $G_1$. Next, at moment $T_2$, conversion of the system shall be performed to achieve goal $G_2$. The system will stop functioning at moment $T_3$. Because $q$ varies, a separate CTS and its functioning model are built for each $q$ but only one model is considered in the example. We need all models ($q = 15$). Furthermore, we need to measure $w_{ju}, p_{ju}$ depending on the possible IT characteristics used to perform $A_0^i \ldots A_3^i$ and so depending on this information, technological operations characteristics are used to create appropriate prescriptions for conversion and further target operations. These IT characteristics are $\{I_a : a = \overline{1, A}\}$. Depending on $I_a$ and $\mathbf{S}_q^r$ different models $M_{qa}$ are built, and as a result, different $s_{uj}$ are realized. So once all these models are built, we will add $q$ and $a$ indexes to the results of the modeling and will get a multidimensional array (tensor) $\mathbf{W}_{[Q,A,J,U]}$, which allows obtaining $w_{qaju}, p_{qaju}$ for each $\mathbf{S}_q^r$ in $\mathbf{S}^r$ and $I_a$ in $\mathbf{I}$:

$$\mathbf{W}_{[Q,A,J,U]} = <<< w_{qaju}, p_{qaju}, u = \overline{1, U} >, j = \overline{0, J} >, q = \overline{1, Q}, a = \overline{1, A} > . \tag{1}$$

Dimensions of (1) are evaluated based on computations of partial network fragment states. The computations are made based on the functional dependencies between the states caused by

the execution of network operations. They can be represented as, for example, computation graphs allowing the computation of the dimensions of (1). The example of a complex technical system's model used to estimate (1) is shown in Figure 1.

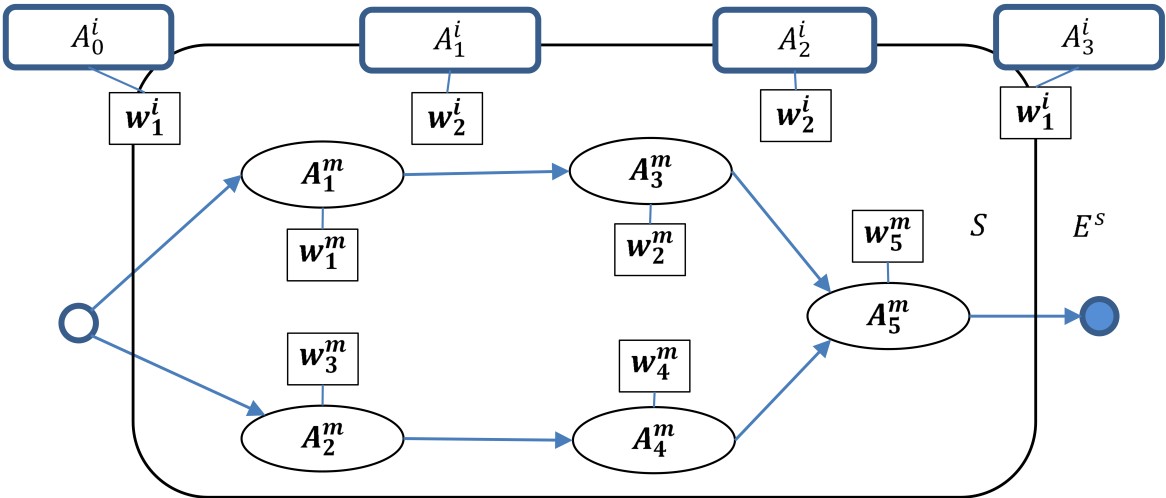

**Figure 1.** An example of the model of a complex technical system.

In this Figure, $w_1^m - w_5^m$ are workplaces to perform the technological noninformation operations (TNIO or material technological operations); $w_1^i, w_2^i$ are workplaces to perform technological information operations (TIO); $A_1^m - A_5^m$ are technological noninformation operations; $A_1^i, A_2^i$ are technological information operations; $S$ is the system; $E^s$ is the environment of the system; and $\hat{t}$ is a random value $t$. Routes used to move elements through CTS's workplaces are shown in Figure 2. In the example routes are similar to the technological operations sequences because of one technological operation allowed on each workplace.

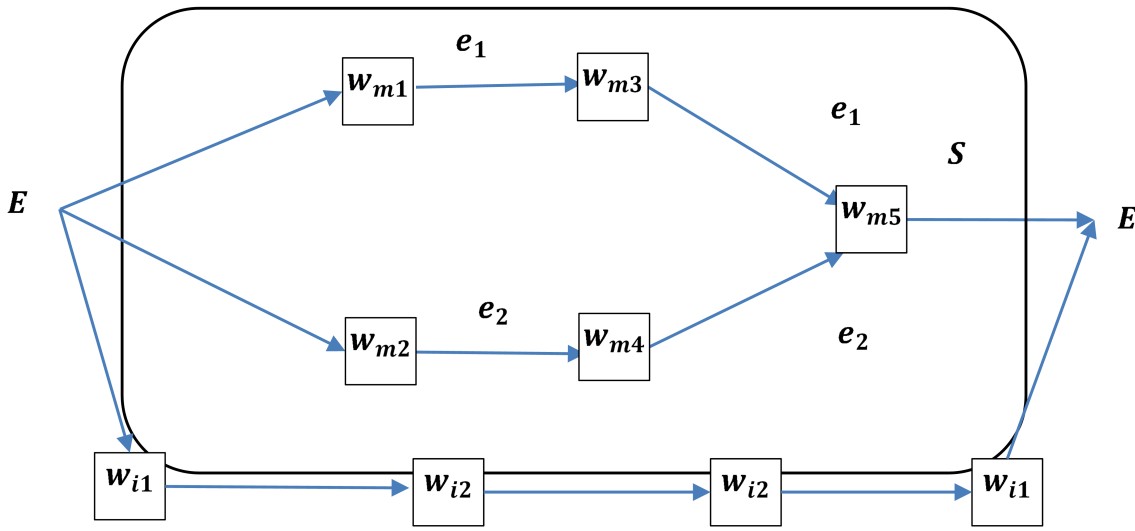

**Figure 2.** The example of information and noninformation technological routes of a complex technical system.

The parameters of the information technological operations (designated by the upper index $i$), are characterized by the left and right margins of the respective random values, so they represented as vectors of random values parameters:

$$A_0^i \sim < t_0^i, c_0^i >; t_0^i = < 1, 3 >; c_0^i = < 1, 2 >; A_1^i \sim < t_1^i, c_1^i >; t_1^i = < 1, 4 >; c_1^i = < 2, 3 >;$$

$$A_2^i \sim < t_2^i, c_2^i >; t_2^i = < 1,2 >; c_2^i = < 1,3 >; A_3^i \sim < t_3^i, c_3^i >; t_3^i = < 1,3 >; c_3^i = < 1,2 >;$$

The calendar schedule of the technological noninformation operations has a matrix form:

$$<< A_1^m, T_1 >, < A_2^m, T_2 >, < A_3^m, T_3 >, < A_4^m, T_4 >, < A_5^m, T_5 >> .$$

Here, $T_c, c = \overline{1,5}$ is the calendar moment of the $A_c^m$ beginning. The technological route shown in Figure 2 has the form of one information route $w_1^i - w_2^i - w_1^i$ and two noninformation route chains: The route chain $w_1 - w_3 - w_5$ is to produce and assemble part 1, and the route chain $w_2 - w_4 - w_5$ is to produce and assemble part 2. The parts are assembled at workplace number 5 ($w_5$). The required states of the $\mathbf{S}_{23}$ sequence, for which the complex technical system model was built, are as follows: $S_{00}^r$ the initial state checked at $T_0 = 0 : \hat{C}_{00}^r = 0$, $\hat{R}_{00} = 0; S_{01}^r$ - first state checked at $\mathbf{T}_1$ : $\hat{C}_{01}^r, \hat{R}_{01} = < \hat{R}_{011}, \hat{R}_{013} >$ . After checking, the goal is changed to $G_1$, so the planned conversion and the required states are changed accordingly. $S_{12}^r$ is the second state checked at $\mathbf{T}_2 : \hat{C}_{12}^r, \hat{R}_{12} = < \hat{R}_{121}, \hat{R}_{123}, \hat{R}_{122}, \hat{R}_{124} >$ . After checking, the goal is changed to $G_2$, and the planned conversion and the required states are changed accordingly. $S_{23}^r$ is the third state checked at $T_3 : \hat{C}_{23}^r, \hat{R}_{23} = < \hat{R}_{235} >$ . A network of technological noninformation (material) operations with delays is shown in Figure 3.

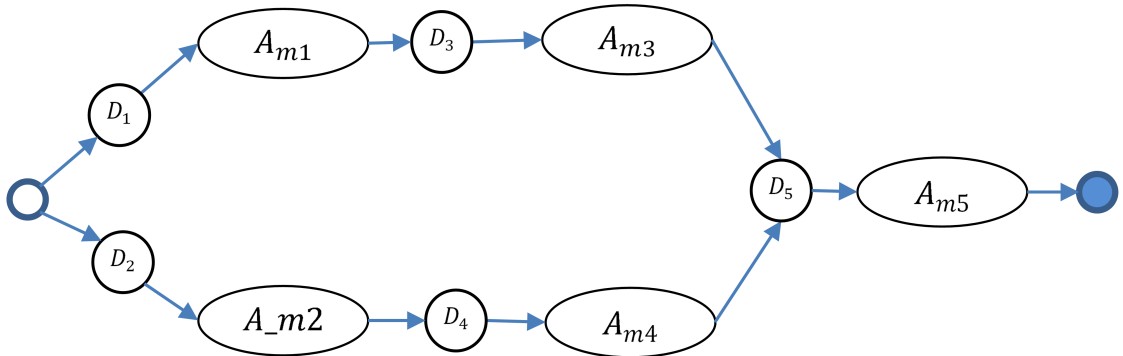

**Figure 3.** An example of a technological operation network with delays.

This is a typical representation of operations—for example, for project management—but expanded. More precisely, the model of CTS functioning is expanded with the use of waiting operations $D_1, D_2, D_3, D_4, D_5$. Waiting operations are used, particularly, to account for calendar plan fulfillment and to represent possible waits (delays) and states of workplaces during each moment of CTS functioning. Next, this model is expanded with a chain of technological information operations in the network specified. The technological information operations are performed in workplaces $w_1^i, w_2^i$. Workplace $w_1^i$ is used to receive and send reports from/to the environment and is not capable of altering CTS functioning, but workplace $w_2^i$ is capable of altering CTS functioning and is not used to receive/send reports to/from the environment. A network of technological information and noninformation operations with waits is shown in Figure 4.

The CTS model created allows the specification of $\mathbf{S}_j-$ which is the fixed set of possible CTS states under condition vector $S_{ij}^r$ of states, required by the CTS environment at $T_j$. Each CTS state is associated with the $b_s^s - s-$th branch at tree $T_{ij}$ of possible branches of simultaneously performing technological operations. It is created for fixed $S_{ij}^r$. Each branch $b_s^s$ is associated with a set of $U$ information and noninformation technological operations and waits for operations, $b_s^s \sim A_{us}, u = \overline{1, U}$, each one performed (or delayed) at workplace $w_u$. This allows the computation of the states of CTS, which correspond to $b_s^s$. Such a tree $T_{ij}$ fragment is shown in Figure 5. The fragment is built under the condition that the technological information operation is performed. Other fragments have the same structure and correspond to cases in which other technological information operations are performed. As a result, a complex tree is built based on its fragments formed. It is shown in Figure 6.

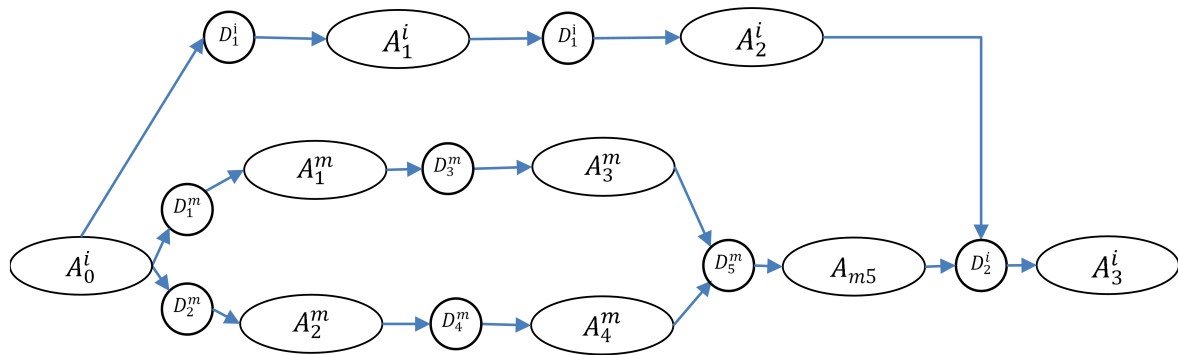

**Figure 4.** An example of a network with technological operations with delays and technological information operations.

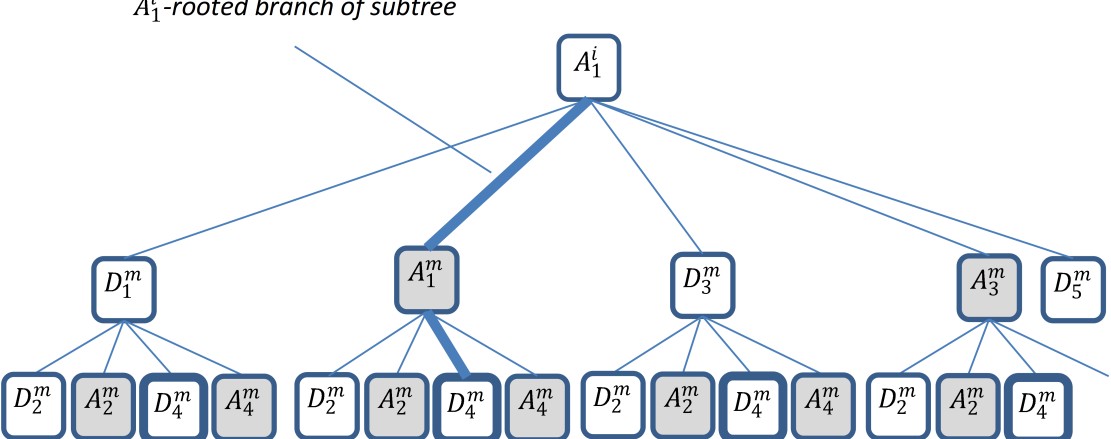

**Figure 5.** A fragment of a tree of initial technological operation network cuts.

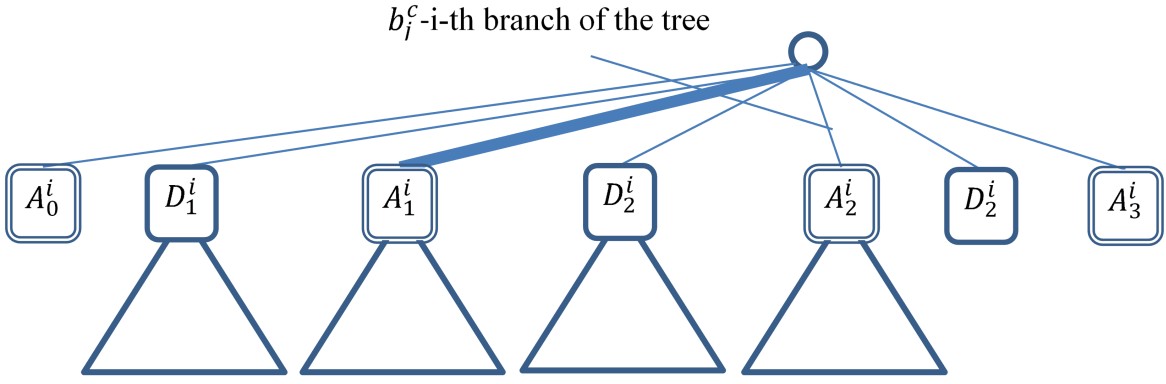

**Figure 6.** Complex tree.

The trees built allow the specification and subsequent computation of the possible states of CTS during its functioning to reach the requirements specified by $S_{ij}^r$. The computation is performed based on the ability to form states of CTS based on the states of $w_u \sim A_{us}$. Let us designate such a state of CTS as $S_s^s \sim b_s^s$. Furthermore, the models of possible states allow us to generate correct conversion technological operations and further technological operations in case the requirements were changed. More precisely, the conversion operations depend on state $S_s^s \sim b_s^s$ and the states required by the environment when CTS is in this state. One partial state of the example of a partial tree is shown in bold in Figure 5. The corresponding complex tree (built with partial ones) is shown in Figure 6. The example state corresponds to the implementation of the $D_1^i$ wait and $A_{m1}$ operation at the start

of checking the compliance of the system with the environmental states and the $D_4$ wait to start $A_{m4}$ technological noninformation operation. In this case, the conversion measures consist of bringing workplace $w_1$ to its original state and then to execute the conversion (readjustment) of $w_1$ to reach the new requirements according to the new goal $G_1$. For $w_4$, it is necessary to perform a readjustment. The information technological operation $A_1^i$ for the purpose of conversion must return as a result of information about the network of conversion technological operations to perform and a calendar plan for their implementation, as well as a network of further technological operations to reach $G_1$ ("target" operations) and a calendar plan for their implementation. Such a situation of alternating functioning due to environmental change, corresponding information operation results, and further conversion and a target operation start is named cutting. An example is shown in Figure 7.

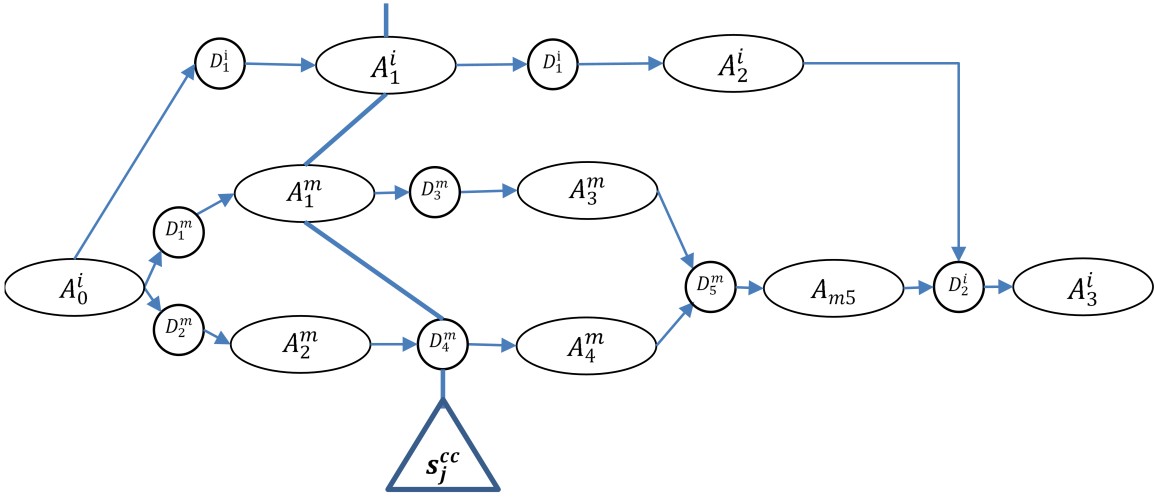

**Figure 7.** An example of cutting.

For simplicity, conversion technological operations cannot be interrupted. In addition, it is supposed that before a new network of operations will be performed, all conversion operations shall end. After the information technological operation $A_2^i$ has finished and the required information obtained the conversion, target operations should start on the third and fourth workplaces according to the specified calendar plan. These target technological operations can also be interrupted when the next state check is performed at $T_2$. The corresponding example of conversion technological operations is shown in Figure 8. They end with technological information operation $A_3^i$ which checks a new state of workplaces and starts "target" technological operations. The corresponding network of technological information and noninformation operations with waits is shown in Figure 9. Please note that this network is generated under a few conditions: first, under the condition of $S_{qij}^r$ fixed with $q = 9$, $\mathbf{S}_2$ and $S_{23}$, next, under the condition of $I_a$ fixed; as a result, $S_u^s$ is fixed, which is built for $< A_1^i, A_1^m, D_4^m >$; and so under the condition of $I_a$ the $A_1^i$ characteristics are fixed. This fixation causes an appropriate $M_{aq}$ and, next, networks of conversion and target technological operations.

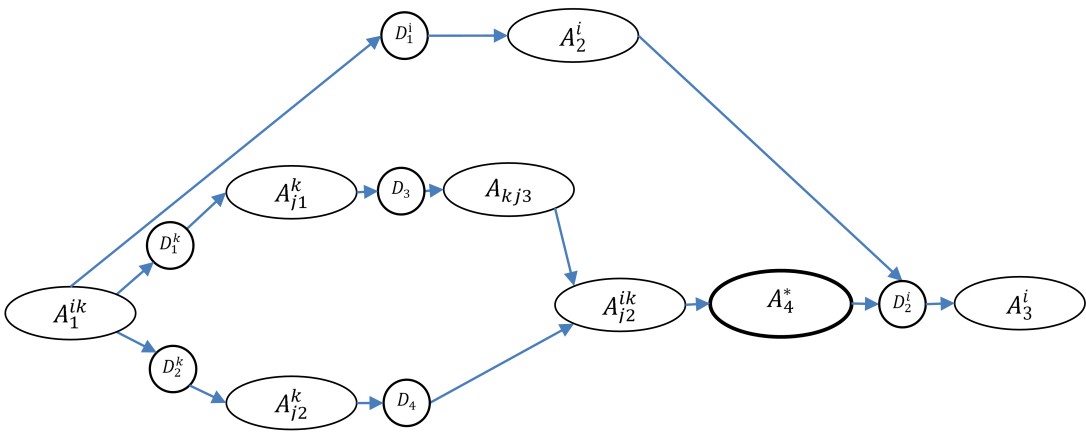

**Figure 8.** An example of a network of conversion operations.

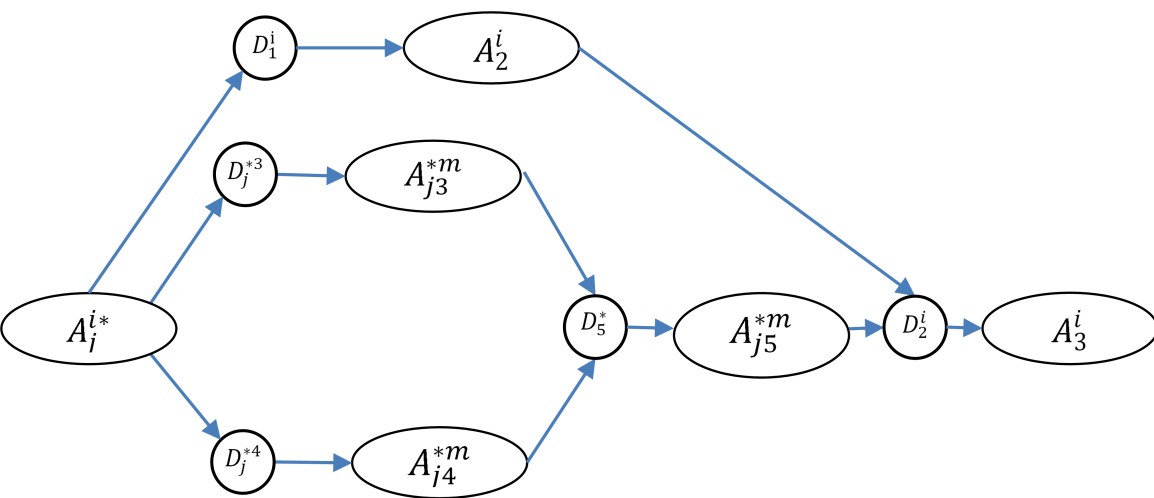

**Figure 9.** An example of a network of technological operations after conversion.

The fulfillment of the new target network of technological operations, according to the required states $\mathbf{S}_{23}$ is interrupted again at $T_2$. Let us consider an example corresponding to the implementation of the CTS state $s_h^s(s_u^s)$ from the set of possible states $\mathbf{S}_h^s(s_u^s)$. Such states are again obtained by the same routine to build an appropriate tree of possible technological operations and waits at CTS functioning after the interruption in specified conditions of the CTS environment $\mathbf{S}_q$ and after the technological information operation with characteristics defined by the specified IT $I_a$. The set of possible states is determined by such operation results, i.e., these are prescriptions (plans, orders) assigned for execution when $A_4^i$ has finished, which depends on the state of CTS during the performing of technological information operations. So, the states of CTS during alternated functioning depend on the previous states of the CTS, as well as on the states of the environment. The appropriate tree of possible branches of simultaneously performing technological operations after the first interruption is shown in Figure 10.

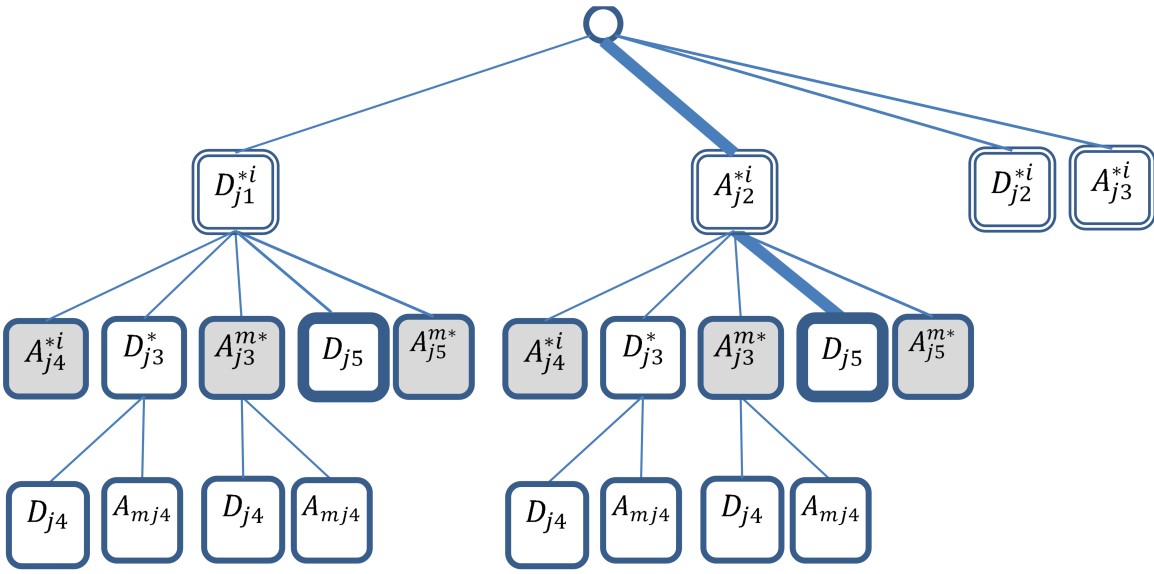

**Figure 10.** Tree of technological operation network cuts after conversion operations.

The example case of the second interruption at $T_2$ (shown in bold) corresponds to the implementation (at moment $T_2$ when the system and environmental states are checked for compliance) of the following technological operations: waiting $D_{j5}$ for the start of the assembly of the $A_{j5}^m$ technological operation (the last technological noninformation operation in the calendar plan). It is assumed that in this case, conversion technological operations are not needed; i.e., the set of conversion technological operations is empty. That is because the technological operation on the fifth workplace $w_5$ has not started yet, and the remaining places have already been restored to their original state. Therefore, $w_1 - w_4$ are supposed to be no longer needed to achieve the new goal $G_2$. In this case, the information operation $A_{hk1}^i$ for the purpose of conversion should immediately return an empty set after its start; next, it shall call the information operation $A_{h2}^i$ which prepares the beginning of the target technological operations. The $A_{h2}^i$ information operation should end immediately (since the state did not change), and the $A_{h5}^m$ (new, according to goal $G_2$) product assembly technological operation should start (probably, with wait $D_{h5}$) on the fifth workplace $w_5$. It is assumed that this technological operation is no longer interrupted. After it has completed the final information technological operation $A_{h4}^i$, it starts without waiting. The appropriate (final) network of technological operations is shown in Figure 11.

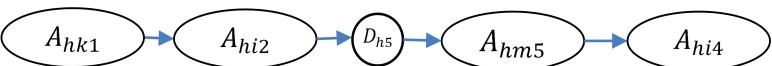

**Figure 11.** Last part of the operation network after the second interruption and the second conversion operations.

As a result, of the series of conditional states and their changes due to the effects of information and noninformation technological operations, i.e., random moments $\hat{T}_u$ of the realization of states $s_u$, the costs $\hat{C}_u$) spent to realize states $s_u$ and the numbers $\hat{\mathbf{R}}_u$ of the parts produced can be computed. Next, the $\hat{T}, \hat{C}$ random variables and the random vector $\hat{\mathbf{R}}$ of the produced items can be computed for each CTS functioning, represented by $M_{qu}$. Such a computation is made based on the assumption that $\hat{T}, \hat{C}$ are distributed according to a beta distribution and that $\hat{\mathbf{R}}$ distributed is distributed according to a binomial distribution. It is further assumed that for large enough networks, the distribution of the resulting effects form a Gauss distribution for $\hat{T}, \hat{C}$ and a binomial distribution for $\hat{\mathbf{R}}$. As a result, the $s_u$ values can be computed. A pair of models, $M_q^e, M_{qa}^s$ is obtained. Next, based on this pair of models, the model $M_{qa}^{se}$ of CTS states and states of its environmental correspondence is created. It depends

on the functioning of the CTS environment according to $S_q^e$ and IT $I_a$ used. The number of such pairs corresponds to $Q_A$. Next, a multidimensional array (1) of $p_{qaju}, w_{qaju}$ values can be evaluated. The values of $p_q$ are determined by the model of the CTS environment. It is worth noticing that the environment model may vary for the IT used and the type of environments (e.g., combat environments, information attack environments, supportive environments), but that is not modeled in this example. This model as a multidimensional array can serve as a comprehensive indicator of the CTS potential regarding the IT $I_a$ used. This array, in fact, sets a multidimensional discrete random vector distribution. The vector elements are probabilities $w_{qaju}$ of the compliance of the random effects with the changing requirements of the CTS environment. Furthermore, this discrete random vector distribution may solve the comprehensive indicator of capabilities, organizational capabilities, and dynamic capabilities of the CTS. However, to solve the practical problems of CTS potential research based on mathematical models (e.g., mathematical programming or operation research models), scalar indicator preferable. It is possible to use many or a few characteristics of this multidimensional discrete random vector or some kind of probabilistic mix as a scalar CTS potential indicator $\psi_1(I_a)$ as a function of IT $I_a$. For example,

$$\psi_1(I_a) = \sum_{q=1}^{Q} (\prod_{j=1}^{J} \sum_{u=1}^{U} (w_{qaju}(I_a, \mathbf{S}_q) p_{qaju}(I_a, \mathbf{S}_q))) p_q; \tag{2}$$

where $w_{qaju}, p_{qaju}$ in (2) are taken from $\mathbf{W}_{[Q,A,J,U]}$ in (1). Alternatively, the CTS potential indicator $\psi_2(I_a)$ can be computed as guaranteed (pessimistic) value:

$$\psi_2(I_a) = \sum_{q=1}^{Q} (\prod_{j=1}^{J} \min_{\forall u \in \overline{1,U}} (w_{qaju}(I_a, \mathbf{S}_q)) p_q; \tag{3}$$

In any case, information technology $I_a$ performance indicator $\Phi(I_a, I_0)$ can be compared to a basic—for example, not digital—IT $I_0$ and can be estimated as a difference:

$$\Phi_1(I_a, I_0) := \psi_1(I_a) - \psi_1(I_0), \text{ or, alternatively } \Phi_2(I_a, I_0) := \psi_2(I_a) - \psi_2(I_0). \tag{4}$$

Equations (1)–(4) can be used, similarly, to estimate indicators of capabilities, organizational capabilities, and dynamic capabilities for CTS functioning in changing environments.

## 4. Discussion

As a result of the research it is possible to overcome the existing gap between the need to solve problems in operational properties (particularly research on system potential and dynamic capability indicators) regarding digitalization based on mathematical models and methods and the lack of the necessary concepts and methodology for solving such problems. Examples of such problems are the optimal usage of distributed ledger technologies for business processes, robotic technological process optimization, and choosing cyberphysical system characteristics. Further research should allow the estimation of indicators of capabilities, organizational capabilities, and dynamic capabilities for CTS functioning in changing environments regarding IT use and the estimation of indicators depending on the characteristics of the environments—for example, combat environments, information attack environments, and supportive and collaborative environments. For this reason, we are planning to describe typical digital transformation research problems with corresponding mathematical problems statements of system's potential research. Possible problem statements include choosing the best information operations, choosing IT and information operation characteristics for the optimal implementation of new IT, and choosing the best digitalization scenarios. Suggested indicators can be used, similarly, to estimate indicators of capabilities, organizational capabilities, and dynamic capabilities for CTS functioning in changing environments.

## 5. Conclusions

The results obtained enable the evaluation of the predicted values of the operational properties of systems (system potential, dynamic capability, organizational capability, IT capability) depending on the IT used for system functioning changes in changing conditions. Corresponding IT-usage indicators, dynamic capabilities, or system potential indicators can be estimated as functional dependencies from variables and parameters. Analytical estimation of such indicators becomes possible depending on the variables and parameters in the mathematical problems to be solved. This could lead to a solution to contemporary problems in research using predictive analytical mathematical models and mathematical methods. Examples of such research problems are problems related to IT productivity and efficiency and the estimation, analysis, and synthesis of the dynamic capabilities of systems. Further research planned to study the modeling, computational and automation aspects of mentioned practical and corresponding them mathematical problems decision.

**Author Contributions:** Conceptualization, methodology, I.L.; formal analysis, investigation, models, writing, A.G. All authors have read and agreed to the published version of the manuscript.

**Funding:** The reported study was funded by RFBR, project number 20-08-00649 and 19-08-00989.

**Conflicts of Interest:** The authors declare no conflict of interest.

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
