# Peer review of "System Potential Estimation with Regard to Digitalization: Main Ideas and Estimation Example"

_information, doi:10.3390/info11030164_

Round 1

Reviewer 1 Report

I suggest to rewrite the abstract and indicate more precisely the necessary information about the article (topicality, purpose, methods used (detailed "mathematical methods"), main results, limitations of the research) and try to avoid the use of the same words in one sentence or two successive sentences.

The same with the keywords - some of them could be excluded. 

Please check the sentences on row 2-3; 27; 92-93.

I suggest revising English as there are difficulties to understand the idea of the sentence (row 41-42; 105-106 "its research" hows?; 118-119 "proposed by author" there are two authors, 185-186, 187, etc.)

It is necessary to refer to Figure 3 in the text. It is and advisable to put each figure in the text as close to its description as possible.

In row 243 there is a reference to Figures 6 and 7, but the word "figure" is missing. It is better to add for better understanding for the reader.

Summarizing I can conclude that the authors present their practical part and conclusions better than the Abstract, Introduction and Theoretical background (part 2). Therefore I suggest revising not only the Abstract (as indicated above) but rethinking and clarify the Introduction and theoretical part to make it more clear for the reader; highlight the most important ideas and reasonably present. 

The article covers an interesting and important topic; however, its topicality should be outlined more clearly. There are no hypotheses identified; the methods and tools should be described more accurately. In some places (abstract, introduction, theoretical part), it isn't easy to understand English.

Author Response

Article was revised. 

Abstract, introduction rewritten, hypothesis of the research added, all deficiences mentioned corrected. 

Article was sent for proofreading to native speakers and updated accordingly.

Reviewer 2 Report

The paper has numerous shortcomings. First of all, its goal is unknown. Reading the paper we are no able to find it in any of its parts. In this situation it is hard to understand what goals Authors wanted to achieve and assess if they succeeded in this. Also the research methods have not been presented and described by the Authors. The quality of English language is also very poor. It happens that reading some of the sentences one is not able to understand its meaning. Also readability of the paper is very poor. In fact it’s hard to imagine the segment of readers for whom this paper could be interesting.

Author Response

Article was rewritten. Deficiences mentioned was corrected.

Article was sent for proofreading by native speakers and corrected accordingly. 

Round 2

Reviewer 1 Report

I would suggest reading the article once more to the authors. They indicated that native speaker revises the article; however there are still some places that are difficult to understand, sometimes there are two or three same words in one sentence or one after another sentence. Maybe it will help reading the text after acceptance of trach changes.

The Introduction is too long; it covers a theoretical background while the theoretical part is sufficiently detailed.

The Abstract also could be shortened and carefully read;  revised English - this is the business card of the article.

Please check the "vectors" presented on page 7 and 8 and their description.

What is the difference between Figure 2 and Figure3? Check numeration of figures, including the text.

After acceptance of trach changes, maybe there will be a possibility to rearrange the presentation of figures trying to put them as near as possible to their description in the text - for better understanding for the reader.

Summarising I would suggest to shorten (if possible, sometimes it seems that information is repeating) the theoretical part and to extend the analytical with discussions. The discussion part is missing.

Author Response

Abstract rewritten,

introduction shortened, final part expanded.  

7-9th pages corrected, Figures 2 and 3 difference explained. 

Figures rearranged. 

Added discusson part.

Reviewer 2 Report

The paper is now acceptable for publication.

Author Response

Thank you